

# Alleles of *CYP3A5* and their association with renal function in chronic kidney disease

Onnapa Kongphan[1], Worachart Lert-itthiporn[2,3,4], Ubon Cha'on[2,3], Sirirat Anutrakulchai[2,5], Kanokwan Nahok[2,3], Nadthanicha Artkaew[2,3], Chanpen Sriphan[6] and Apinya Jusakul[2,7]

[1] Department of Biomedical Science, Faculty of Graduate School, Khon Kaen University, Khon Kaen, Thailand
[2] Chronic Kidney Disease Prevention in the Northeast of Thailand (CKDNET) Project, Khon Kaen University, Khon Kaen, Thailand
[3] Department of Biochemistry, Faculty of Medicine, Khon Kaen University, Khon Kaen, Thailand
[4] Center for Translational Medicine, Faculty of Medicine, Khon Kaen University, Khon Kaen, Thailand
[5] Department of Internal Medicine, Faculty of Medicine, Khon Kaen University, Khon Kaen, Thailand
[6] Srinagarind Hospital, Faculty of Medicine, Khon Kaen University, Khon Kaen, Thailand
[7] The Centre for Research and Development of Medical Diagnostic Laboratories, Faculty of Associated Medical Sciences, Khon Kaen University, Khon Kaen, Thailand

Corresponding author
Apinya Jusakul, apinjus@kku.ac.th

## ABSTRACT

**Background:** The cytochrome P450 family 3 subfamily A polypeptide 5 (*CYP3A5*) gene plays an important role in renal function through its product's involvement in metabolizing endogenous substances and drugs, including immunosuppressants used following kidney transplantation. A single-nucleotide polymorphism, *CYP3A5*\*3 (rs776746), produces a non-functional variant that may influence progression of chronic kidney disease (CKD) by impairing renal filtration. However, the frequency of the *CYP3A5*\*3 allele in the Thai population and its association with renal parameters remain underexplored. This study aimed to determine the prevalence of *CYP3A5* polymorphisms and their association with renal function.
**Methods:** We investigated the distribution of *CYP3A5* polymorphisms in 329 northeastern Thai participants, including 205 CKD patients and 124 healthy controls. Genotyping was performed using the TaqMan allelic discrimination assay. Renal function parameters were assessed and compared between *CYP3A5*\*1 and *CYP3A5*\*3 allele carriers.
**Results:** In the entire cohort, the allele frequency of *CYP3A5*\*3 was 63.2%, with genotype frequencies of *CYP3A5*\*1/\*1 (16.7%), *CYP3A5*\*1/\*3 (40.1%), and *CYP3A5*\*3/\*3 (43.2%). There was no significant difference in the *CYP3A5* allele frequencies between CKD and control groups. *CYP3A5*\*3 carriers exhibited significantly lower eGFR, urine creatinine and serum creatinine clearance and higher UACR compared to *CYP3A5*\*1 carriers. After adjusting for confounders, *CYP3A5*\*3 remained significantly associated with reduced urine creatinine.
**Conclusion:** This study highlights a high prevalence of *CYP3A5* polymorphisms in the northeastern Thai population. The association of the *CYP3A5*\*3 allele with renal function parameters underscores the need for further research into the mechanisms

by which *CYP3A5* affects kidney function, which could inform personalized CKD management strategies.

## INTRODUCTION

Chronic kidney disease (CKD) is a significant global health issue and a major cause of mortality and morbidity worldwide, particularly among the population in the northeast region of Thailand (*Jager et al., 2019*; *Cha'on et al., 2022*). Additionally, there has been a steady increase in the number of patients receiving kidney transplants in Thailand (*Larpparisuth, Cheungpasitporn & Lumpaopong, 2021*). Besides the major risk factors such as diabetes and hypertension that lead to CKD progression, genetic variation may also contribute to CKD risk (*Kazancioğlu, 2013*; *Obrador et al., 2017*; *Cha'on et al., 2022*). Currently, CKD management generally relies on pharmacological therapy, such as antihypertensive and antidiabetic medicines, to manage renal function and prevent renal failure. Despite significant advances in pharmacological interventions the challenge remains to develop more personalized treatments that can more effectively prevent CKD progression (*Ruggenenti, Cravedi & Remuzzi, 2012*; *Chen, Knicely & Grams, 2019*).

Among the genetic factors potentially influencing CKD outcomes is the gene encoding cytochrome P450 family 3 subfamily A member 5 (*CYP3A5*) (*Lee et al., 2021*). *CYP3A5* polymorphisms, particularly the *CYP3A5*\*3 allele, are common loss-of-function variants that affect *CYP3A5* protein activity (*Kuehl et al., 2001*). The *CYP3A5* gene, part of the cytochrome P450 family on chromosome 7q21.1, encodes an enzyme that metabolizes various endogenous substrates, including steroids, lipids and arachidonic acid metabolites, as well as certain clinical drugs, including immunosuppressants used in kidney transplant patients (*Willrich et al., 2008*; *Zanger & Schwab, 2013*; *Chen & Prasad, 2018*). Among the known alleles (*CYP3A5*\*1 through *9), *CYP3A5*\*3 is the most prevalent, with its frequency varying across populations (*Kuehl et al., 2001*; *Rodriguez-Antona et al., 2022*). This allele (6986A>G, rs776746) results in a non-functional protein due to aberrant mRNA splicing, leading to significantly reduced or absent CYP3A5 enzyme activity (*Kuehl et al., 2001*). The CYP3A5 enzyme plays an important role in homeostatic signaling mechanisms related to kidney function including the metabolism of steroids, regulation of mineralocorticoid signaling, and modulation of the renin–angiotensin–aldosterone system (RAAS) (*Lidberg et al., 2021*). CYP3A5 can also modulate sodium retention and oxidative stress production in renal proximal tubules, which may contribute to kidney injury (*Koch et al., 2002*; *Lidberg et al., 2021*). Additionally, CYP3A5 metabolizes arachidonic acid derivatives, including 19-hydroxyeicosatetraenoic acid (19-HETE) and 6β-hydroxycortisol, both of which are involved in renal vascular regulation (*Knights, Rowland & Miners, 2013*). The loss of CYP3A5 function due to the *CYP3A5*\*3 allele is associated

with altered 19-HETE metabolism, which may impact renal hemodynamics and contribute to the progression of CKD (*Afshinnia et al., 2020*). Understanding the role of *CYP3A5* polymorphisms is therefore critical for developing genetically informed CKD treatment strategies.

Several studies have reported that frequency of the *CYP3A5*\*3 allele is higher in Asian populations (77% in Japan, 76% in China, 61% in Malaysia, and 59% in India) than among African-Americans (48%) (*Kuehl et al., 2001*; *Fukuen et al., 2002*; *Balram et al., 2003*). Furthermore, previous studies have shown varying results regarding the association between *CYP3A5*\*3/\*3 and CKD progression across different ethnicities. In Malaysia population, the *CYP3A5*\*3 allele has been reported at high prevalence of 53% in CKD patients with adverse drug reaction of antihypertensive therapy (*Lee et al., 2022*). Additionally, the *CYP3A5*\*3/\*3 allele has been associated with rapid CKD progression, defined as an eGFR decline of more than 5 mL/min/1.73 m$^2$ per year (*Lee et al., 2021*). In cases of severe CKD progression leading to end-stage renal disease (ESRD), the prevalence of *CYP3A5*\*3/\*3 allele was 49.59% among Indian renal transplant recipients, who required a lower dose of tacrolimus, a key immunosuppressive drug, compared to those with *CYP3A5*\*1/\*1 (prevalence of 12.5%) (*Prasad et al., 2020*). Similarly, in the Thai population, the *CYP3A5* genotype-guide tacrolimus dosing has been shown to improve therapeutic target achievements, reinforcing the significance of pharmacogenomics in CKD management (*Anutrakulchai et al., 2019*).

Beyond immunosuppressants, *CYP3A5* polymorphisms may also impact the metabolism of other drugs commonly used in CKD treatment, such as antihypertensive agents. For instance, amlodipine, a widely used calcium channel blocker, is primarily metabolized by CYP3A5. Studies have shown that individuals with the *CYP3A5*\*3/\*3 genotype exhibit altered amlodipine clearance, which may affect blood pressure control and drug efficacy (*Huang et al., 2017*; *Liang et al., 2021*). Given the significant role of CYP3A5 in renal drug metabolism, its polymorphisms may influence therapeutic outcomes in CKD patients and should be considered in personalized medicine approaches.

These findings suggest that early detection of *CYP3A5* genetic variants may aid in tailoring drug therapy and predicting CKD progression, ultimately helping to reduce the risk of ESRD. Thus, this study aimed to investigate the prevalence of the *CYP3A5*\*3 allele in the Thai population and in individuals with CKD, and to assess the association between *CYP3A5*\*3 carriers and renal function parameters.

# MATERIALS AND METHODS

## Participants and study definitions

This retrospective case-control study comprised 329 participants, including 205 CKD patients and 124 healthy controls. Samples, demographic and clinical data were derived from participants who were enrolled in the projects "Chronic Kidney Disease Prevention in the Northeast of Thailand (CKDNET) screening program" between 2017–2019 (*Cha'on et al., 2020*, *2022*) and "Genetic profiling in Chronic Kidney Disease patients in Khon Kaen: targeting for diagnostic and prognostic biomarkers" in 2023–2024. Written informed consent was obtained from all participants. This study was approved by the

Khon Kaen University Ethics Committee for Human Research (HE641252). Participants were recruited from northeastern Thailand, primarily from Khon Kaen province, with a gender distribution of 57.8% in female and 42.2% in male, and a mean age of was 60.4 years. Inclusion criteria for the CKD group followed criteria established by the Kidney Disease: Improving Global Outcomes (KDIGO) work group (*KDIGO CKD Work Group, 2013*). Participants were aged ≥18 years with either of the following criteria for more than 3 months: (1) estimated glomerular filtration rate (eGFR) <60 ml/min/1.73 m$^2$, or (2) presence of any kidney damage; urine albumin-creatinine ratio (UACR) ≥30 mg/g in two consecutive samples, and presence of urine red blood cells ≥3–5 cells/high power field in two consecutive samples (*Cha'on et al., 2022*). CKD staging was classified based on eGFR level with stage 1; ≥90, stage 2; 60–89, stage 3a; 45–59, stage 3b; 30–44, stage 4; 15–29, and stage 5; <15 ml/min/1.73 m$^2$. The UACR was used to define 3 groups including A1; <30, A2; 30–300, and A3 >300 mg/g. Control participants were defined as individuals aged ≥18 years with eGFR ≥60 ml/min/1.73 m$^2$, who did not have a clinical diagnosis of CKD (*Cha'on et al., 2022*). To ensure the exclusion of individuals with early kidney disease, participants with diabetes, hypertension, or UACR >30 mg/g were not included in the control group. Additionally, all control participants underwent repeat eGFR and UACR assessments over a 3-month period, and those with persistent eGFR <60 ml/min/1.73 m$^2$ or evidence of structural kidney abnormalities were excluded, following KDIGO diagnostic criteria.

Among the variables of interest, hypertension was defined as a persistent systolic blood pressure (SBP) ≥140 mmHg, and diastolic blood pressure (DBP) ≥90 mmHg or on the basis of diagnosis by physicians and use of antihypertensive drugs. Diabetes was defined by levels of hemoglobin A1c (HbA1c) ≥6.5% or based on diagnosis by physicians and use of diabetes medication. Anemia was defined by hemoglobin levels <13 g/dL in males and <12 g/dL in females. Serum creatinine clearance was calculated using the Cockcroft-Gault equation: creatinine clearance = [140 – age (years)] * weight (kg)]/[72 * serum creatinine (mg/dL)], adjusted by multiplying by 0.85 for women (*Froissart et al., 2005*).

## Sample size

The sample size for this study was estimated to determine the association between *CYP3A5* and CKD using G*Power (*Faul et al., 2007*). To ensure population specificity, genetic background data for the Thai population were incorporated. A systematic review by *Dorji, Tshering & Na-Bangchang (2019)* reported that the prevalence of the *CYP3A5*\*3/\*3 genotype in the Thai population was 37.17%, based on data from studies conducted in Bangkok and central Thailand (*Dorji, Tshering & Na-Bangchang, 2019*). The sample size calculation based on an odds ratio (OR) of 4.190 (*Lee et al., 2021*), a significance level (alpha) of 0.05, and a desired power of 0.80. The odds ratio was converted to Cohen's h using the formula: h = 2 × (arcsin($\sqrt{p1}$)) – (arcsin($\sqrt{p0}$)), where p1 represents the prevalence of the polymorphism in cases and p0 in controls. Sample size was calculated based on statistically independent t-test. The calculated sample size was 34 cases and 34 controls with a power of 80%. Given that our study included a substantially larger sample size (205 CKD patients and 124 controls), this confirms that our study was

adequately powered to detect significant associations between *CYP3A5*\*3 polymorphisms and renal function parameters.

## Genotyping by TaqMan allelic discrimination

Genomic DNA (gDNA) was extracted from whole blood using the QIAamp DNA Mini Kit (Qiagen, Hilden, Germany). *CYP3A5* polymorphism was determined using TaqMan Drug Metabolism Genotyping assay and TaqMan Genotyping Master Mix (Applied Biosystems, Foster City, CA, USA). The assay used two sequence-specific probes (assay ID: C__26201809_30): 5′-ATGTGGTCCAAACAGGGAAGAGATA[T]TGAAAGACAAA AGAGCTTTTAAAG-3′ labeled with VIC fluoroprobe, and 5′-ATGTGG TCCAAACAGGGAAGAGATA[C]TGAAAGACAAAAGAGCTCTTTAAAG-3′ labeled with FAM fluorophore. When a single nucleotide variant, either A (*CYP3A5*\*1 genotype) or G (*CYP3A5*\*3) binds to the probe, the signal of VIC or FAM was detected, respectively. Cycling reactions were performed using a CFX96 Real-Time PCR System (Bio-Rad, Hercules, CA, USA) with the following conditions: 95 °C for 10 min followed by 50 cycles of 95 °C for 15 s, and 60 °C for 90 s. Allelic discrimination analysis was conducted using the CFX Maestro software (Version 2.3; Bio-Rad, Hercules, CA, USA). To ensure accuracy and reproducibility, several quality control measures were implemented. No-template control (NTC) was used as a negative control, and gBlocks Gene Fragments of known *CYP3A5*\*1/\*1, \*1/\*3, and \*3/\*3 genotypes were used as positive controls. To assess the reliability and consistency of the genotyping results, the samples were genotyped in duplicate within the same run. Furthermore, direct sequencing was performed on randomly selected samples representing all three genotypes to validate TaqMan assay results.

## Statistical analyses

Data are expressed as frequencies for categorical variables, mean ± standard deviation (SD) for normally distributed numerical variables, and median with interquartile ranges (IQR) for non-normally distributed numerical variables. The chi-square test was used to assess Hardy–Weinberg equilibrium. Clinical and demographic data of CKD and control subjects was compared using chi-square for categorical variables. An independent t-test and Mann-Whitney U test were used to compare normally and non-normally distributed numerical data, respectively. The renal function and clinical characteristics were compared using one-way ANOVA tests for normally distributed numerical variables or Kruskal-Wallis tests for non-normally distributed numerical variables, when stratified by *CYP3A5* genotype distribution.

To evaluate the relationship between *CYP3A5* alleles and renal function parameters, simple and multivariate linear regression analyses were performed. Renal function parameters, including eGFR, UACR, serum creatinine clearance, and urine creatinine levels, were selected as dependent variables based on statistically significant differences ($p < 0.05$) between *CYP3A5*\*1 and *CYP3A5*\*3 allele carriers, as identified in univariate analysis. Although CKD staging is commonly used to categorize kidney function, the renal parameter, particularly eGFR focused on continuous variables provides greater statistical

power and sensitivity to detect subtle differences in renal function, particularly in early disease stages. This approach is aligned with previous genetic studies investigating renal traits (*Köttgen et al., 2009*; *Lee et al., 2021*). Potential confounding factors were selected based on their well-established influence on renal function and CKD progression. Age, gender, hypertension, diabetes status, and CKD status were included as independent variables in the multivariate regression models. Age was considered due to its direct effect on renal function decline, while gender differences influence creatinine metabolism and eGFR. Hypertension and diabetes were included as major risk factors for CKD progression, and CKD status was considered to account for differences in renal function between CKD patients and healthy controls. Simple linear regression was initially performed to assess the unadjusted associations between *CYP3A5*3* allele carriers and renal function parameters. Multivariate linear regression was subsequently conducted, incorporating all factors that were statistically significant ($p < 0.05$) in the simple linear regression analysis, as well as the predefined confounding variables. All statistical analyses were conducted using SPSS Statistics, version 28.0 (IBM Corporation; Armonk, NY, USA), with statistical significance defined as $p < 0.05$.

## RESULTS

### Characteristics of the study population

The demographic and genotyping data are shown in Table 1 and Table S1. In total, 329 participants were included: 205 CKD and 124 control subjects. The mean age of the study population was 60.4 ± 11.3 years: those with CKD had a higher mean age than controls (63.4 ± 11.4 *vs.* 55.4 ± 9.0 years, respectively; $p < 0.001$). Compared with the control group, the CKD patients had significantly higher levels of SBP, FPG, HbA1C, serum creatinine, and UACR, along with significantly lower eGFR and serum creatinine clearance ($p < 0.001$). There was no significant difference in BMI, DBP, and urine creatinine between the two groups.

### Distribution of the *CYP3A5* polymorphism

The genotype frequencies of *CYP3A5*3* followed Hardy-Weinberg equilibrium assumptions in both the CKD ($p = 0.595$) and control groups ($p = 0.614$), indicating a stable allele distribution within the studied population. The overall genotype frequencies were *1/*1 (A/A) at 16.7%, *1/*3 (A/G) at 40.1%, and *3/*3 (G/G) at 43.2% (Table 1).

Among individuals with CKD, the genotype frequencies were 17.6% (*CYP3A5*1/*1*), 38.0% (*CYP3A5*1/*3*), and 44.4% (*CYP3A5*3/*3*), while in the control group, the distributions were 15.3%, 43.5%, and 41.2%, respectively. Importantly, no significant difference in genotype frequency was observed between the CKD and control groups ($p = 0.605$). However, renal function parameters showed differences between CKD and control groups. CKD patients exhibited significantly lower eGFR ($p < 0.001$), higher UACR ($p < 0.001$), and higher serum creatinine ($p < 0.001$) compared to controls. This finding suggesting that *CYP3A5*3* is not directly associated with CKD susceptibility.

When comparing the *CYP3A5*3* allele frequency (63.2%) in the northeastern Thai population with global populations, we found that it is lower than in East Asians

**Table 1 Demographics and clinical characteristics of the study population.**

| Characteristics | Entire cohort (n = 329) | CKD (n = 205) | Control (n = 124) | p-value |
|---|---|---|---|---|
| Age (years)[a] | 60.4 ± 11.3 | 63.4 ± 11.4 | 55.4 ± 9.0 | **<0.001** |
| Female, n (%) | 190 (57.8) | 103 (50.2) | 87 (70.2) | **<0.001** |
| Male, n (%) | 139 (42.2) | 102 (49.8) | 37 (29.8) | |
| BMI (kg/m$^2$)[a] | 24.3 ± 4.1 | 24.1 ± 4.2 | 24.8 ± 4.0 | 0.158 |
| SBP (mmHg)[b] | 129 (118–142) | 137 (124–149) | 119 (112–129) | **<0.001** |
| DBP (mmHg)[a] | 78.1 ± 10.8 | 78.5 ± 12.4 | 77.3 ± 7.5 | 0.297 |
| FPG (mg/dl)[b] | 90 (83–102) | 94 (84–111.8) | 87 (80.3–93.8) | **<0.001** |
| HbA1c(%)[b] | 5.7 (5.3–6.1) | 5.8 (5.4–6.3) | 5.5 (5.3–5.8) | **<0.001** |
| eGFR (ml/min/1.73 m$^2$)[b] | 72.3 (41.8–89.1) | 51.7 (27.0–73.6) | 87.3 (77.6–93.6) | **<0.001** |
| **CKD stage, n (%)** | | | | |
| 1 (eGFR ≥90 ml/min/1.73 m$^2$) | 77 (23.4) | 27 (13.2) | – | |
| 2 (eGFR 60–89 ml/min/1.73 m$^2$) | 132 (40.1) | 58 (28.3) | – | |
| 3a (eGFR 45–59 ml/min/1.73 m$^2$) | 30 (9.1) | 30 (14.6) | – | |
| 3b (eGFR 30–44 ml/min/1.73 m$^2$) | 32 (9.7) | 32 (15.6) | – | |
| 4 (eGFR 15–29 ml/min/1.73 m$^2$) | 30 (9.1) | 30 (14.6) | – | |
| 5 (eGFR <15 ml/min/1.73 m$^2$) | 28 (8.5) | 28 (13.7) | – | |
| Serum creatinine (mg/dL)[b] | 0.99 (0.80–1.59) | 1.25 (0.93–2.31) | 0.82 (0.74–0.92) | **<0.001** |
| Serum creatinine clearance (ml/min)[b] | 59.1 (30.7–76.8) | 38.0 (26.1–61.8) | 75.0 (66.6–86.9) | **<0.001** |
| **Albuminuria, n (%)** | | | | |
| A1 (<30 mg/day) | 213 (64.7) | 89 (3.4) | 124 (100.0) | **<0.001** |
| A2 (30–300 mg/day) | 53 (16.1) | 53 (25.9) | – | |
| A3 (>300 mg/day) | 63 (19.1) | 63 (30.7) | – | |
| UACR (mg/g)[b] | 10.2 (5.0–115.2) | 48.8 (7.1–532.1) | 6.0 (4.1–8.4) | **<0.001** |
| Urine creatinine (mg/dL)[b] | 86.5 (56.8–141.0) | 86.2 (59.4–139.9) | 89.8 (54.6–142.6) | 0.814 |
| *CYP3A5 genotypes (rs776746), n (%)* | | | | |
| *1/*1 (A/A) | 55 (16.7) | 36 (17.6) | 19 (15.3) | 0.605 |
| *1/*3 (A/G) | 132 (40.1) | 78 (38.0) | 54 (43.5) | |
| *3/*3 (G/G) | 142 (43.2) | 91 (44.4) | 51 (41.2) | |
| *CYP3A5 alleles, n (%)* | | | | |
| CYP3A5*1 | 242 (36.8) | 150 (36.6) | 92 (37.1) | 0.895 |
| CYP3A5*3 | 416 (63.2) | 260 (63.4) | 156 (62.9) | |

**Notes:**
[a] Variables are presented as mean ± SD and compared using independent t-test.
[b] Variables are presented as median (interquartile ranges) and compared using Mann–Whitney U tests. Unless otherwise stated, all variables are presented as frequencies and compared using the chi-square test. Bold values indicate statistically significant p-values (p < 0.05).
BMI, body mass index; SBP, systolic blood pressure; DBP, diastolic blood pressure; FPG, fasting plasma glucose; HbA1C, hemoglobin A1C; eGFR, estimated glomerular filtration rate; UACR, urine albumin-creatinine ratio.

(72.2–77.5%), South Asians (74.9%), and Europeans (92.9%) but higher than in Africans (30.3%) (Table 2). This pattern aligns closely with prior studies in other Southeast Asian populations, such as Malaysians and Indonesians, reinforcing the genetic background of CYP3A5*3 in this region.

**Table 2 Comparison of *CYP3A5\*3* (rs776746 A>G) allele frequency in different populations.**

| CYP3A5*3 (rs776746) | Frequency (%) | References |
|---|---|---|
| Global | 88.7 | ALFA |
| European | 92.9 | ALFA |
| Korean | 77.5 | ALFA and Korea1K |
| Chinese | 76.0 | *Balram et al. (2003)* |
| Japanese | 75.0 | ALFA and ToMMo 14KJPN |
| South Asian | 74.9 | ALFA |
| East Asian | 72.2 | ALFA |
| Other Asian | 68.0 | ALFA |
| Northeastern Thai | 63.2 | This study |
| African | 30.3 | ALFA |
| African Others | 17.1 | ALFA |

Note:
   ALFA, allele frequency aggregator; Korea1K, the Korean genome project including 1,094 whole genomes; ToMMo 14KJPN; genotype frequency panel provides genotype frequency information on SNVs and INDELs detected by short-read WGS analysis of 14,000 Japanese individuals.

## The association between *CYP3A5* allele and renal function parameters in the entire cohort

We next determined the association between the *CYP3A5\*3* allele and renal function parameters (Table 3). Among these, age, SBP, eGFR, UACR, urine creatinine, and serum creatinine clearance were significantly different when comparing individuals with one or two *CYP3A5\*3* alleles to those without ($p = 0.010$, $p = 0.015$, $p = 0.047$, $p = 0.003$, $p = 0.023$ and $p = 0.030$, respectively). Specifically, in individuals with a *CYP3A5\*3* allele, the level of UACR was significantly higher ($p = 0.003$), while eGFR, urine creatinine, and serum creatinine clearance were significantly lower compared to those with the *CYP3A5\*1* allele ($p = 0.047$, $p = 0.023$ and $p = 0.030$, respectively).

When stratified by genotype distribution, the differences in renal function parameters remain evident. Participants with *CYP3A5\*1/\*3* and *CYP3A5\*3/\*3* genotypes had lower urine creatinine ($p = 0.017$) and serum creatinine clearance levels ($p = 0.062$), indicating that the *CYP3A5\*3* allele may impair creatinine filtration efficacy. UACR levels were elevated in both *CYP3A5\*1/\*3* and *CYP3A5\*3/\*3* genotypes when compared with the *CYP3A5\*1/\*1* wild type genotype ($p = 0.018$), indicating a higher degree of albuminuria in individuals with a *CYP3A5\*3* allele. However, no significant difference was observed in urine microalbumin levels across the different genotype groups, suggesting that *CYP3A5* polymorphisms only impact certain aspects of renal function, and may not significantly alter overall glomerular filtration rates.

To further investigate these associations, we performed simple linear and multivariate regression analyses to identify independent predictors of renal function parameters (Table 4). In simple regression, the presence of the *CYP3A5\*3* allele was significantly associated with lower urine creatinine levels ($\beta = -12.520$, $p = 0.011$), lower eGFR ($\beta = -5.098$, $p = 0.030$), and lower serum creatinine clearance ($\beta = -4.828$, $p = 0.047$). However, after adjusting for confounding factors including age, gender, hypertension, and

**Table 3** Renal function and clinical characteristics stratified by *CYP3A5* allele and genotype distribution in the entire cohort.

| Variables | Allele (*n* = 658) | | | Genotype (*n* = 329) | | | |
|---|---|---|---|---|---|---|---|
| | *CYP3A5*1* (*n* = 242) | *CYP3A5*3* (*n* = 416) | *p*-value | *CYP3A5*1/*1* (*n* = 55) | *CYP3A5*1/*3* (*n* = 132) | *CYP3A5*3/*3* (*n* = 142) | *p*-value |
| Female, *n* (%) | 143 (59.1) | 237 (57.0) | 0.596 | 32 (58.2) | 79 (59.8) | 79 (55.6) | 0.778 |
| Male, *n* (%) | 99 (40.9) | 179 (43.0) | | 23 (41.8) | 53 (40.2) | 63 (44.4) | |
| Age (years)[a] | 58.9 ± 11.1 | 61.2 ± 11.3 | **0.010** | 56.5 ± 10.3 | 60.9 ± 11.4 | 61.4 ± 11.2 | **0.018** |
| BMI (kg/m$^2$)[a] | 24.1 ± 3.7 | 24.5 ± 4.3 | 0.220 | 24.4 ± 3.2 | 23.8 ± 4.1 | 24.8 ± 4.4 | 0.140 |
| SBP (mmHg)[b] | 126 (117–141) | 131 (119–144) | **0.015** | 123 (117–140) | 130 (118–141) | 131 (119–146) | 0.072 |
| DBP (mmHg)[a] | 78.6 ± 10.3 | 77.7 ± 11.1 | 0.302 | 80.3 ± 9.3 | 77.3 ± 10.9 | 77.9 ± 11.2 | 0.227 |
| eGFR (ml/min/1.73 m$^2$)[b] | 74.3 (47.9–90.7) | 71.0 (37.7–88.3) | **0.047** | 74.3 (48.1–93.3) | 74.0 (43.7–88.1) | 67.6 (36.7–88.4) | 0.175 |
| UACR (mg/g)[b] | 7.1 (3.8–63.0) | 10.9 (4.8–187.8) | **0.003** | 6.9 (3.7–41.6) | 11.6 (5.5–116.0) | 11.1 (5.1–186.7) | **0.018** |
| Urine creatinine (mg/dL)[b] | 91.7 (60.1–151.3) | 82.9 (6.4–130.2) | **0.023** | 115.7 (64.3–178.9) | 85.4 (51.4–132.4) | 82.6 (59.0–129.4) | **0.017** |
| Urine microalbumin (mg/dL)[b] | 0.9 (0.4–6.2) | 1.1 (0.4–28.8) | 0.074 | 0.9 (0.4–4.3) | 0.9 (0.4–11.3) | 1.3 (0.4–36.2) | 0.247 |
| Serum creatinine (mg/dL)[b] | 1.0 (0.8–1.3) | 1.0 (0.8–1.7) | 0.156 | 1.0 (0.8–1.3) | 1.0 (0.8–1.5) | 1.0 (0.8–1.8) | 0.407 |
| Serum creatinine clearance (ml/min)[b] | 61.7 (34.7–81.1) | 55.6 (29.9–75.2) | **0.030** | 65.1 (44.0–86.6) | 57.9 (30.7–75.0) | 55.1 (29.6–75.7) | 0.062 |
| CKD, *n* (%) | 150 (62.0) | 260 (62.5) | 0.895 | 36 (65.5) | 78 (59.1) | 91 (64.1) | 0.605 |
| **Albuminuria status, *n* (%)** | | | | | | | |
| A1 (<30 mg/day) | 167 (69.0) | 259 (62.3) | 0.064 | 40 (72.7) | 87 (65.9) | 86 (60.6) | 0.134 |
| A2 (30–300 mg/day) | 40 (16.5) | 66 (15.9) | | 11 (20.0) | 18 (13.6) | 24 (16.9) | |
| A3 (>300 mg/day) | 35 (14.5) | 91 (21.9) | | 4 (7.3) | 27 (20.5) | 32 (22.5) | |

**Notes:**
[a] Variables are presented as mean ± SD. Allele frequencies were compared using independent t-tests, and genotypes compared using one-way ANOVA.
[b] Variables are presented as median (interquartile ranges) and allele frequencies compared using Mann–Whitney U tests, and genotypes compared using Kruskal-Wallis tests. Unless otherwise stated, all variables are presented as frequencies and compared using chi-square tests. Bold values indicate statistically significant *p*-values (*p* < 0.05). BMI, body mass index; SBP, systolic blood pressure; DBP, diastolic blood pressure; eGFR, estimated glomerular filtration rate; UACR, urine albumin-creatinine ratio.

diabetes status, only urine creatinine levels remained significantly associated with *CYP3A5*3* ($\beta = -11.390$, $p = 0.018$), while eGFR ($\beta = -2.182$, $p = 0.213$), serum creatinine clearance ($\beta = -0.684$, $p = 0.702$), and UACR levels ($\beta = 63.895$, $p = 0.395$) were no longer significantly associated with *CYP3A5*3*.

Overall, these results support the hypothesis that *CYP3A5* loss-of-function variants may impair renal filtration efficiency. The *CYP3A5*3* allele was significantly associated with reduced urine creatinine levels and lower eGFR, indicating potential renal dysfunction. Additionally, UACR levels were elevated in *CYP3A5*3* carriers, suggesting an increased risk of albuminuria and glomerular damage.

## The association between *CYP3A5* allele and renal function parameters in CKD patients

To evaluate the association between the *CYP3A5*3* allele and renal function in CKD patients, key renal parameters were compared between individuals with and without the *CYP3A5*3* allele (Table 5). Significant differences were observed in age, SBP, eGFR, UACR, urine microalbumin, serum creatinine, and serum creatinine clearance between

**Table 4 Factors associated with renal parameters including eGFR, urine creatinine, serum creatinine clearances and UACR levels in the entire cohort.**

| Predictors | Simple linear regression | | | Multivariate linear regression | | |
|---|---|---|---|---|---|---|
| | β | SE | p-value | β | SE | p-value |
| **eGFR levels** | | | | | | |
| CYP3A5 allele (*1 vs *3) | −5.098 | 2.348 | **0.030** | −2.182 | 1.750 | 0.213 |
| Age (Years) | −1.067 | 0.092 | **<0.001** | −0.475 | 0.081 | **<0.001** |
| Gender (Male vs Female) | 11.487 | 2.256 | **<0.001** | 4.578 | 1.734 | **0.008** |
| Hypertension (No vs Yes) | −25.526 | 2.337 | **<0.001** | −9.396 | 2.147 | **<0.001** |
| Diabetes (No vs Yes) | −36.941 | 2.482 | **<0.001** | −21.577 | 2.333 | **<0.001** |
| CKD (No vs Yes) | −33.379 | 1.949 | **<0.001** | −17.761 | 2.170 | **<0.001** |
| **Urine creatinine levels** | | | | | | |
| CYP3A5 allele (*1 vs *3) | −12.520 | 4.892 | **0.011** | −11.390 | 4.818 | **0.018** |
| Age (Years) | −0.211 | 0.213 | 0.322 | | | |
| Gender (Male vs Female) | −16.092 | 4.768 | **<0.001** | −18.560 | 4.736 | **<0.001** |
| Hypertension (No vs Yes) | −14.309 | 5.193 | **0.006** | −16.400 | 5.906 | **0.006** |
| Diabetes (No vs Yes) | −16.781 | 5.842 | **0.004** | −11.312 | 5.212 | **0.030** |
| CKD (No vs Yes) | 3.133 | 4.958 | 0.528 | | | |
| **Serum creatinine clearances levels** | | | | | | |
| CYP3A5 allele (*1 vs *3) | −4.828 | 2.423 | **0.047** | −0.684 | 1.789 | 0.702 |
| Age (Years) | −1.460 | 0.086 | **<0.001** | −1.0226 | 0.084 | **<0.001** |
| Gender (Male vs Female) | 10.913 | 2.338 | **<0.001** | 4.362 | 1.775 | **0.014** |
| Hypertension (No vs Yes) | −22.267 | 2.396 | **<0.001** | −7.689 | 2.139 | **0.014** |
| Diabetes (No vs Yes) | −31.374 | 2.600 | **<0.001** | −14.784 | 2.328 | **<0.001** |
| CKD (No vs Yes) | −29.697 | 2.140 | **<0.001** | −12.386 | 2.235 | **<0.001** |
| **UACR levels** | | | | | | |
| CYP3A5 allele (*1 vs *3) | 138.135 | 83.701 | 0.099 | 63.895 | 75.053 | 0.395 |
| Age (Years) | 14.737 | 3.549 | **<0.001** | −0.253 | 3.475 | 0.942 |
| Gender (Male vs Female) | −145.220 | 81.682 | 0.076 | | | |
| Hypertension (No vs Yes) | 741.025 | 85.662 | **<0.001** | 526.614 | 91.848 | **<0.001** |
| Diabetes (No vs Yes) | 1,039.835 | 93.762 | **<0.001** | 868.150 | 99.906 | **<0.001** |
| CKD (No vs Yes) | 596.656 | 80.140 | **<0.001** | 93.545 | 91.739 | 0.308 |

**Note:**
β, Unstandardized beta; SE, Coefficients standardized error; Hypertension is defined as blood pressure ≥140/90 mmHg or clinical diagnosis of hypertension, Diabetes is defined as HbA1C ≥6.5% or clinical diagnosis of diabetes. Bold values indicate statistically significant p-values ($p < 0.05$).

CYP3A5*3 carriers and non-carriers ($p = 0.031$, $p = 0.004$, $p = 0.008$, $p = 0.007$, $p = 0.011$, $p = 0.020$, and $p = 0.016$, respectively).

CKD patients harboring at least one CYP3A5*3 allele exhibited higher UACR ($p = 0.007$), elevated urine microalbumin levels ($p = 0.011$), and increased serum creatinine ($p = 0.020$), while eGFR ($p = 0.008$) and serum creatinine clearance ($p = 0.016$) were significantly lower compared to non-carriers. These findings suggest that the CYP3A5*3 allele may contribute to reduced renal function and increased albuminuria in CKD patients.

Table 5 Renal function and clinical characteristics stratified by *CYP3A5* allele and genotype distribution in CKD patients.

| Variables | Allele (*n* = 410) | | | Genotype (*n* = 205) | | | |
|---|---|---|---|---|---|---|---|
| | *CYP3A5*1* (*n* = 150) | *CYP3A5*3* (*n* = 260) | *p-value* | *CYP3A5*1/*1* (*n* = 36) | *CYP3A5*1/*3* (*n* = 78) | *CYP3A5*3/*3* (*n* = 91) | *p-value* |
| Female, *n* (%) | 80 (53.3) | 126 (48.5) | 0.342 | 20 (55.6) | 40 (51.3) | 43 (47.3) | 0.682 |
| Male, *n* (%) | 70 (46.7) | 134 (51.5) | | 16 (44.4) | 38 (48.7) | 48 (52.7) | |
| Age (years)[a] | 64.0 (53.0–70.3) | 65.0 (57.0–73.0) | **0.031** | 65.0 (56.0–72.5) | 67.0 (58.8–73.0) | 65.0 (56.0–73.0) | **0.011** |
| BMI (kg/m$^2$)[a] | 23.9 (20.9–26.7) | 23.2 (21.0–26.7) | 0.980 | 24.3 (21.5–27.5) | 23.2 (20.7–25.8) | 23.2 (21.6–27.4) | 0.481 |
| SBP (mmHg)[b] | 134.2 ± 17.7 | 139.4 ± 18.1 | **0.004** | 130.2 ± 15.5 | 137.7 ± 18.9 | 140.1 ± 17.8 | **0.019** |
| DBP (mmHg)[b] | 79.5 ± 11.5 | 78.0 ± 12.9 | 0.213 | 81.0 ± 10.1 | 78.1 ± 12.6 | 77.8 ± 13.1 | 0.408 |
| eGFR (ml/min/1.73 m$^2$)[a] | 60.9 (31.8–78.3) | 47.3 (23.8–70.7) | **0.008** | 68.5 (35.3–81.3) | 54.7 (30.0–72.9) | 45.4 (21.4–66.4) | 0.050 |
| UACR (mg/g)[a] | 35.3 (5.5–272.3) | 78.9 (8.8–793.4) | **0.007** | 11.9 (4.4–120.3) | 63.6 (7.9–909.5) | 68.2 (9.3–801.0) | **0.012** |
| Urine creatinine (mg/dL)[a] | 88.1 (52.6–150.7) | 85.5 (59.7–130.8) | 0.247 | 107.1 (61.5–187.9) | 85.7 (51.3–132.4) | 84.4 (59.8–130.4) | 0.163 |
| Urine microalbumin (mg/dL)[a] | 3.0 (0.8–20.8) | 7.3 (1.0–55.8) | **0.011** | 2.2 (0.8–10.1) | 5.5 (0.7–70.1) | 9.9 (1.1–55.2) | **0.043** |
| Serum creatinine (mg/dL)[a] | 1.1 (0.9–1.9) | 1.3 (0.9–2.5) | **0.020** | 1.1 (0.9–1.6) | 1.2 (0.9–2.0) | 1.4 (1.0–2.8) | 0.101 |
| Serum creatinine clearance (ml/min)[a] | 47.1 (27.8–64.8) | 35.5 (24.6–58.2) | **0.016** | 58.9 (27.6–81.2) | 37.8 (27.7–58.6) | 35.3 (20.5–57.9) | 0.055 |
| CKD, n (%) | 150 (100.0) | 260 (100.0) | – | 36 (100.0) | 78 (100.0) | 91 (100.0) | – |
| **Albuminuria status, *n* (%)** | | | | | | | |
| A1 (<30 mg/day) | 75 (50.0) | 103 (39.6) | **0.036** | 21 (58.3) | 33 (42.3) | 35 (38.5) | 0.078 |
| A2 (30–300 mg/day) | 40 (26.7) | 66 (25.4) | | 11 (30.6) | 18 (23.1) | 24 (26.4) | |
| A3 (>300 mg/day) | 35 (23.3) | 91 (35.0) | | 4 (11.1) | 27 (34.6) | 32 (35.2) | |

Notes:
[a] Variables are presented as median (interquartile ranges) and allele frequencies compared using Mann-Whitney U tests, and genotypes compared using Kruskal-Wallis tests.
[b] Variables are presented as mean ± SD and allele frequencies compared using independent t-tests, and genotypes compared using one-way ANOVA; unless otherwise stated, all variables are presented as frequencies and compared using chi-square tests. Bold values indicate statistically significant *p*-values ($p < 0.05$).
BMI, body mass index; SBP, systolic blood pressure; DBP, diastolic blood pressure; eGFR, estimated glomerular filtration rate; UACR, urine albumin-creatinine ratio.

When stratified by genotype, the differences in UACR and urine microalbumin levels remained significant, with participants carrying the *CYP3A5*1/*3* and *CYP3A5*3/*3* genotypes exhibiting higher UACR ($p = 0.012$) and urine microalbumin levels ($p = 0.043$) compared to *CYP3A5*1/*1* carriers. This further supports the hypothesis that *CYP3A5*3* may exacerbate albuminuria and contribute to renal dysfunction in CKD patients.

To further assess whether *CYP3A5*3* independently contributes to renal dysfunction, we performed simple and multivariate linear regression analyses (Table 6). In simple regression, the *CYP3A5*3* allele was significantly associated with lower eGFR ($\beta = -7.564$, $p = 0.013$), lower serum creatinine clearance ($\beta = -6.363$, $p = 0.038$), higher serum creatinine levels ($\beta = 0.478$, $p = 0.008$), and higher urine microalbumin levels ($\beta = 15.694$, $p = 0.043$). However, after adjusting for age, gender, hypertension, and diabetes status, these associations were no longer statistically significant, suggesting that the observed effects may be largely influenced by these clinical factors rather than a direct effect of *CYP3A5*3* on renal function.

**Table 6 Factors associated with the renal parameters including eGFR, serum creatinine levels, serum creatinine clearances, UACR, and urine microalbumin levels in CKD patients.**

| Predictors | Simple linear regression | | | Multivariate linear regression | | |
|---|---|---|---|---|---|---|
| | β | SE | p-value | β | SE | p-value |
| **eGFR levels** | | | | | | |
| CYP3A5 allele (*1 vs *3) | −7.564 | 3.026 | **0.013** | −3.243 | 2.672 | 0.225 |
| Age (Years) | −0.831 | 0.122 | **<0.001** | −0.583 | 0.115 | **<0.001** |
| Gender (Male vs Female) | 10.341 | 2.892 | **<0.001** | 8.912 | 2.564 | **<0.001** |
| Hypertension (No vs Yes) | −10.583 | 2.909 | **<0.001** | −9.537 | 2.586 | **<0.001** |
| Diabetes (No vs Yes) | −24.970 | 2.918 | **<0.001** | −20.629 | 2.820 | **<0.001** |
| **Serum creatinine levels** | | | | | | |
| CYP3A5 allele (*1 vs *3) | 0.478 | 0.180 | **0.008** | 0.332 | 0.169 | 0.051 |
| Age (Years) | 0.000 | 0.008 | 0.952 | | | |
| Gender (Male vs Female) | −0.925 | 0.168 | **<0.001** | −0.865 | 0.163 | **<0.001** |
| Hypertension (No vs Yes) | 0.356 | 0.175 | **0.042** | 0.366 | 0.163 | **0.026** |
| Diabetes (No vs Yes) | 0.994 | 0.181 | **<0.001** | 0.886 | 0.174 | **<0.001** |
| **Serum creatinine clearances levels** | | | | | | |
| CYP3A5 allele (*1 vs *3) | −6.363 | 3.052 | **0.038** | −1.685 | 2.545 | 0.508 |
| Age (Years) | −1.292 | 0.112 | **<0.001** | −1.108 | 0.110 | **<0.001** |
| Gender (Male vs Female) | 8.905 | 2.920 | **0.002** | 6.903 | 2.439 | **0.005** |
| Hypertension (No vs Yes) | −9.632 | 2.922 | **0.001** | −7.627 | 2.453 | **0.002** |
| Diabetes (No vs Yes) | −21.230 | 2.984 | **<0.001** | −14.049 | 2.680 | **<0.001** |
| **UACR levels** | | | | | | |
| CYP3A5 allele (*1 vs *3) | 216.393 | 129.076 | 0.094 | 101.681 | 120.394 | 0.399 |
| Age (Years) | 9.250 | 5.458 | 0.091 | | | |
| Gender (Male vs Female) | −48.274 | 124.750 | 0.699 | | | |
| Hypertension (No vs Yes) | 559.581 | 122.470 | **<0.001** | 522.800 | 116.458 | **<0.001** |
| Diabetes (No vs Yes) | 890.772 | 127.201 | **<0.001** | 863.998 | 124.616 | **<0.001** |
| **Urine microalbumin levels** | | | | | | |
| CYP3A5 allele (*1 vs *3) | 15.694 | 7.735 | **0.043** | 8.549 | 7.151 | 0.233 |
| Age (Years) | 0.407 | 0.328 | 0.215 | | | |
| Gender (Male vs Female) | −23.073 | 7.391 | **0.002** | −19.899 | 6.871 | **0.004** |
| Hypertension (No vs Yes) | 22.567 | 7.426 | **0.003** | 21.986 | 6.901 | **0.002** |
| Diabetes (No vs Yes) | 58.536 | 7.473 | **<0.001** | 55.652 | 7.362 | **<0.001** |

Note:
β, Unstandardized beta; SE, coefficients standardized error; hypertension is defined as blood pressure ≥140/90 mmHg or diagnosed with hypertension, diabetes is defined as HbA1C ≥ 6.5% or diagnosed with diabetes. Bold values indicate statistically significant p-values ($p < 0.05$).

## DISCUSSION

In this study, we investigated the prevalence of the *CYP3A5*3* allele in the Thai population and its association with renal function, particularly in CKD patients. Our findings suggest that the *CYP3A5*3* allele is very common in Thailand, with an overall frequency of 63.2%, comparable to other East Asian populations. Importantly, we identified significant associations between the *CYP3A5*3* allele and some renal function parameters, including

eGFR, UACR, urine creatinine, and serum creatinine clearance. Our study highlights that the *CYP3A5*\*3 allele may contribute to impaired renal function.

The prevalence of the *CYP3A5*\*3 allele varies significantly across different ethnicities (*Thompson et al., 2004*). We demonstrated that the frequency of *CYP3A5*\*3 was comparable to earlier findings in East Asian populations. These frequencies were, however, higher than those in Africa. Our study found a *CYP3A5*\*3 allele frequency of 63.2% in the Khon Kaen population, which is higher than the 37.17% *CYP3A5*\*3/\*3 genotype prevalence previously reported in Thai individuals from central Thailand (*Dorji, Tshering & Na-Bangchang, 2019*). *Dorji, Tshering & Na-Bangchang (2019)* conducted a systematic review across Southeast and East Asia, reporting *CYP3A5*\*3 allele frequencies ranging from 2.0% to 71.4%, with higher frequencies in East Asians (Japanese, Chinese, and Korean populations) and lower frequencies in African populations. Our study's findings align with the upper range of *CYP3A5*\*3 allele frequencies observed in Southeast Asia but highlight potential regional variation within Thailand (*Dorji, Tshering & Na-Bangchang, 2019*). This suggests that genetic background and regional differences within Thailand may influence *CYP3A5* allele distribution, emphasizing the importance of population-specific studies when investigating genetic risk factors for CKD. In Thailand, where the *CYP3A5*\*3 allele is common, tailoring CKD management to genetic risk factors may improve patient outcomes and reduce the burden of CKD-related complications.

The *CYP3A5*\*3 allele is a loss-of-function variant that leads to reduced or absent CYP3A5 enzyme activity (*Givens et al., 2003*). This decrease in enzyme activity can affect the metabolism of endogenous substances and drugs, including those used in CKD treatment, such as immunosuppressants after kidney transplantation (*Oetting et al., 2016*; *Chen & Prasad, 2018*). We found that the *CYP3A5*\*3 allele was associated with higher levels of UACR and lower levels of eGFR, urine creatinine, and serum creatinine clearance, implying that *CYP3A5*\*3 allele carriers may have an increased risk of kidney damage and impaired renal filtration. These findings are consistent with previous studies that have found associations between *CYP3A5*\*3 and poor renal outcomes, including rapid progression of CKD (*Givens et al., 2003*; *Lee et al., 2021*). Serum creatinine clearance was significantly reduced in healthy African American adults carrying the *CYP3A5*\*3 allele, consistent with our findings (*Givens et al., 2003*). Similarly, a study by *Lee et al. (2021)* found that the *CYP3A5*\*3/\*3 genotype was associated with rapid eGFR decline in Malaysian CKD patients (*Lee et al., 2021*). These results, alongside our findings, suggest that *CYP3A5*\*3 may impair renal dysfunction across multiple ethnic groups by impairing creatinine metabolism and increasing albuminuria. Despite these similarities, some discrepancies exist between our findings and prior research. While we observed a significant association between *CYP3A5*\*3 and renal function parameters in unadjusted analyses, these associations were attenuated after adjusting for confounders such as age, hypertension, and diabetes. This suggests that the effect of *CYP3A5*\*3 on renal function may be modulated by other clinical risk factors, highlighting the importance of considering gene-environment interactions in CKD progression.

The mechanism by which CYP3A5 affects renal function remains incompletely understood. CYP3A5 protein is primarily expressed in the proximal tubule and collecting

duct of the kidney, where it plays a role in sodium transport, mineralocorticoid metabolism, and creatinine excretion (*Bolbrinker et al., 2012*; *Lidberg et al., 2021*). Functionally, CYP3A5 metabolizes glucocorticoids *via* 6β-hydroxylation, enhancing mineralocorticoid receptor signaling, which promotes sodium retention and increased blood pressure, all of which are critical contributors to CKD progression. Additionally, CYP3A5 regulates vitamin D metabolism, specifically shunting the metabolic pathway of 25-hydroxyvitamin D3 toward the 4β-hydroxylation pathway in proximal tubules, with altered signaling potentially contributing to RAAS dysregulation and impaired renal function (*Lidberg et al., 2021*). Thus, the reduced activity due to the *CYP3A5*\*3 allele may impair creatinine tubular secretion, resulting in lower urine creatinine levels and decreased creatinine clearance. Furthermore, our findings indicate that *CYP3A5*\*3 carriers had higher UACR levels, suggesting a potential association between CYP3A5 deficiency and glomerular damage or endothelial dysfunction. The increased albumin leakage observed in *CYP3A5*\*3 carriers may result from altered RAAS activity, a pathway known to be influenced by CYP3A5 activity (*Lidberg et al., 2021*). Further mechanistic studies, particularly those exploring the crosstalk between CYP3A5, vitamin D metabolism, and RAAS, are warranted to validate these observations.

Furthermore, our findings demonstrate that while *CYP3A5*\*3 allele frequency did not differ between CKD patients and controls, CKD patients exhibited significantly worse renal function parameters, including lower eGFR and higher UACR and serum creatinine. This suggests that *CYP3A5*\*3 is not directly associated with CKD onset but may contribute to renal function impairment once CKD is established. These findings are consistent with previous studies that reported *CYP3A5*\*3's impact on creatinine clearance and albuminuria but not on CKD susceptibility (*Givens et al., 2003*; *Lee et al., 2021*). Further analysis revealed that while *CYP3A5*\*3 was significantly associated with renal function parameters in CKD patients, this association was no longer significant after adjusting for confounders such as age, hypertension, and diabetes. This suggests that the impact of *CYP3A5*\*3 on renal function may be modulated by clinical risk factors rather than acting as an independent determinant of CKD. These findings highlight the importance of considering genetic contributions within the context of established CKD rather than in disease susceptibility alone.

The high frequency of the *CYP3A5*\*3 allele found in this study implies that a sizable proportion of CKD patients may be genetically predisposed to poorer renal outcomes. This demonstrates the potential benefits of incorporating genetic screening for *CYP3A5* polymorphisms into routine clinical practice, which would enable more personalized treatment strategies based on individual genetic profiles. The presence of *CYP3A5* polymorphisms can also affect the efficacy and safety of pharmacotherapy in patients with CKD (*Willrich et al., 2008*; *Chen & Prasad, 2018*; *Lee et al., 2021*). In particular, patients with the *CYP3A5*\*3/\*3 genotype may require lower doses of certain medications because their bodies will be less efficient at processing these drugs. This could lower the risk of drug toxicity, but it may also necessitate close monitoring to ensure therapeutic efficacy. A recently published randomized controlled trial studied the clinical utility of genotype-guided tacrolimus dosing in kidney transplantation, comparing the outcomes of

a conventional tacrolimus dose with a genotype-guided approach based on *CYP3A5* genotypes (*Anutrakulchai et al., 2019*). Interestingly, the study showed that genotype-guided dosing significantly increased the proportion of patients achieving therapeutic tacrolimus concentrations with fewer patients experiencing over-therapeutic drug levels—particularly in the *CYP3A5*\*3/\*3 genotype group (*Anutrakulchai et al., 2019*). Our findings support these results, as the *CYP3A5*\*3 allele was associated with altered renal function parameters, suggesting that CYP3A5-deficient individuals may require modified dosing strategies for nephrotoxic drugs such as tacrolimus and amlodipine. Furthermore, *CYP3A5*\*3 polymorphisms have been associated to variability in antihypertensive drug metabolism, particularly for calcium channel blockers like amlodipine, which are frequently prescribed to CKD patients. Studies indicate that *CYP3A5*\*3 carriers exhibit reduced amlodipine clearance, leading to prolonged drug exposure and potentially enhanced blood pressure control (*Huang et al., 2017*; *Liang et al., 2021*). Given that hypertension is a major driver of CKD progression, understanding the impact of *CYP3A5* polymorphisms on antihypertensive response is essential for optimizing treatment outcomes in CKD patients.

Given the high prevalence of the *CYP3A5*\*3 allele in the Thai population and its association with impaired renal parameters, these findings highlight the potential of *CYP3A5* genotyping as a tool for precision medicine in CKD management. Integration of *CYP3A5* genotype screening could assist in early risk stratification for patients predisposed to rapid CKD progression and in optimizing pharmacological therapy. Patients carrying the *CYP3A5*\*3 genotype exhibit significantly reduced metabolic capacity, which affects the clearance of medications metabolized by the CYP3A5 enzyme, such as tacrolimus and certain antihypertensives. The Clinical Pharmacogenetics Implementation Consortium (CPIC) recommends genotype-guided tacrolimus dosing, with *CYP3A5*\*3 carriers requiring reduced doses to avoid drug toxicity while maintaining immunosuppressive efficacy (*Birdwell et al., 2015*). This approach has been validated in Thai transplant patients (*Anutrakulchai et al., 2019*), supporting the clinical relevance of genotype-informed therapy. Furthermore, our findings align with the KDIGO 2022 guidelines, which advocate for individualized therapy based on patient-specific risk factors, including genetic variability. Thus, routine implementation of *CYP3A5* genotyping could help tailor CKD treatment regimens, improve drug safety, and reduce the burden of adverse drug reactions, thereby supporting a more personalized and effective CKD care strategy.

This study has several limitations that should be acknowledged. First, the retrospective design may introduce selection bias, as the participants were selected from a specific region in Northeast Thailand and may not fully represent the broader Thai population or other ethnic groups. Additionally, while the overall the study's sample size was sufficient to detect associations, some subgroup analyses may have been underpowered, potentially leading to increased variability in statistical results and limiting the ability to detect more subtle genetic effects or interactions with environmental factors. Further studies with larger, well-stratified cohorts are needed to ensure the stability and reproducibility of these findings. Another limitation is the use of a single SNP (*CYP3A5*\*) to determine genetic associations with CKD. Although this variant is well-studied, other alleles of the *CYP3A5*

gene or other genes involved in renal function could also contribute to CKD risk and progression. Furthermore, the study did not account for potential confounding variables such as diet, medication adherence, or other lifestyle factors that might influence renal function. Finally, while our findings suggest an association between *CYP3A5*\*3 polymorphisms and renal function, the cross-sectional nature of the study precludes conclusions about causality. Longitudinal studies are needed to confirm these associations and to better understand the mechanistic pathways involved. Additionally, while CKD staging and UACR classifications were defined to characterize the study population, they were not directly analyzed for associations with *CYP3A5* polymorphism. Given that CKD stage is inherently determined by eGFR levels and UACR is a categorical measure of albuminuria severity, our study prioritized analyzing renal function parameters (*e.g.*, GFR, creatinine clearance, and UACR as a continuous variable) to assess the impact of *CYP3A5*\*3. This avoids redundancy and provides a more precise evaluation of the role of *CYP3A5*\*3 in kidney function.

## CONCLUSIONS

In this study, we investigated the prevalence of the *CYP3A5*\*3 polymorphism in the northeastern Thai population and its association with renal function, particularly among CKD patients. The study reveals a high frequency (63.2%) of the *CYP3A5*\*3 allele in this population, consistent with other East and Southeast Asian populations. Importantly, we identified significant associations between the *CYP3A5*\*3 allele and key renal function parameters, including eGFR, UACR, urine creatinine, and serum creatinine clearance. These findings indicate that carriers of the *CYP3A5*\*3 allele may be at a higher risk for kidney damage and impaired renal filtration, highlighting the potential role of this polymorphism in CKD progression. The high prevalence of the *CYP3A5*\*3 allele in the Thai population emphasizes the importance of genetic factors in CKD management and the potential for personalized treatment strategies based on genetic profiling. These findings add to the growing body of evidence on the impact of *CYP3A5* polymorphisms on renal function and call for additional research to determine the underlying mechanisms by which CYP3A5 affects kidney function. Additionally, multi-ethnic studies are necessary for further investigation to enhance the understanding of *CYP3A5* polymorphism in relation to renal function and to support the personalized treatment strategies tailored to diverse populations.

## ACKNOWLEDGEMENTS

The authors express gratitude to all participants involved in this study. We thank Kunha Muisuk for technical support. We would like to acknowledge Prof. David Blair, for editing the manuscript *via* Publication Clinic KKU, Thailand. We acknowledge the Centre for Research and Development of Medical Diagnostic Laboratories (CDML). During the preparation of this work the authors used ChatGPT and QuillBot to enhance the language in certain sections of the text. After using this tool, the authors reviewed and edited the content as needed and take full responsibility for the content of the publication.

### Funding

This work was supported by the Fundamental Fund of Khon Kaen University and has received funding support from the National Science Research and Innovation Fund to Apinya Jusakul. Onnapa Kongphan was supported by the Research Fund for Supporting Lecturer to Admit High Potential Student to Study and Research on His Expert Program Year 2022 (651JH103) from Graduate School, Khon Kaen University, Thailand. The funders had no role in study design, data collection and analysis, decision to publish, or preparation of the manuscript.

### Grant Disclosures

The following grant information was disclosed by the authors:
Fundamental Fund of Khon Kaen University.
National Science Research and Innovation Fund.
Research Fund for Supporting Lecturer to Admit High Potential Student to Study and Research on His Expert Program Year 2022: 651JH103.
Graduate School, Khon Kaen University, Thailand.

### Competing Interests

The authors declare that they have no competing interests.

### Author Contributions

- Onnapa Kongphan conceived and designed the experiments, performed the experiments, analyzed the data, prepared figures and/or tables, authored or reviewed drafts of the article, and approved the final draft.
- Worachart Lert-itthiporn conceived and designed the experiments, analyzed the data, authored or reviewed drafts of the article, and approved the final draft.
- Ubon Cha'on conceived and designed the experiments, authored or reviewed drafts of the article, and approved the final draft.
- Sirirat Anutrakulchai conceived and designed the experiments, analyzed the data, authored or reviewed drafts of the article, and approved the final draft.
- Kanokwan Nahok performed the experiments, authored or reviewed drafts of the article, and approved the final draft.
- Nadthanicha Artkaew performed the experiments, authored or reviewed drafts of the article, and approved the final draft.
- Chanpen Sriphan performed the experiments, authored or reviewed drafts of the article, and approved the final draft.
- Apinya Jusakul conceived and designed the experiments, performed the experiments, analyzed the data, prepared figures and/or tables, authored or reviewed drafts of the article, and approved the final draft.

## Human Ethics

The following information was supplied relating to ethical approvals (*i.e.*, approving body and any reference numbers):

The Khon Kaen University Ethics Committee for Human Research.

## Data Availability

The raw data is available in the Supplemental File.

## Supplemental Information

Supplemental information for this article can be found online at http://dx.doi.org/10.7717/peerj.19424#supplemental-information.

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
