# Peer review of "Alleles of *CYP3A5* and their association with renal function in chronic kidney disease"

_PeerJ, doi:10.7717/peerj.19424_

## Round 0.1 · original submission · Major Revisions

Find below the reviewers' comments.

Reviewer 1 ·

Basic reporting

Before discussing the association between CYP3A5 and CKD, the article should provide a detailed background on the field, especially regarding the prevalence of CYP3A5 in other ethnic groups and its role in renal function. This section should include sufficient literature references to establish the background of the study. For example, in addition to the existing references, some new studies can be added, especially those related to the association between CKD, drug metabolism and CYP3A5. When discussing the association between CYP3A5 and CKD, be sure to cite the latest literature in the field, especially those that directly contribute to the study of renal function. Make sure that all information in the article is centered around the research hypothesis and avoid introducing irrelevant discussions. Each section should have a clear goal and directly link to the research hypothesis or question. For example, when discussing the role of CYP3A5 alleles, it should be clearly stated how the study will test the hypothesis that there is an association between the CYP3A5*3 allele and impaired renal function in patients with CKD. Linking results to hypothesis: In the results section, make sure that each experimental result directly answers the research question. For example, when describing the effect of the CYP3A5*3 allele on renal function parameters such as eGFR and UACR, it is necessary to clarify how the results support or deny the research hypothesis. Discussion and hypothesis verification: The discussion section should focus on analyzing the consistency or difference between the research results and the original hypothesis, and compare them with previous studies in the literature to clarify the innovation of this study and its contribution to existing research.

Experimental design

The representativeness of the sample needs to be described in more detail. Although the article mentioned that the participants were from northeastern Thailand, it did not specify whether the regional distribution, gender ratio and other characteristics of the sample were representative to support the extrapolation of the research conclusions. The background related to gene frequency was insufficient. The CYP3A5*3 allele frequency cited in the sample size calculation came from other studies, but it was not stated whether these data were consistent or similar to the Thai population. A discussion on the genetic background specific to the Thai population should be added. CKD staging and UACR grouping were not associated with the experimental objectives. Although the CKD staging criteria and UACR grouping were defined, it was not further explained how these variables were linked to the association study of CYP3A5 alleles in subsequent analysis, which made the connection between the background and the experimental objectives slightly insufficient. The statistical analysis lacked a basis for variable selection. In the multivariate linear regression analysis, it was not clear which variables were potential confounders or key influencing factors, which may affect the interpretation of the results. It is recommended to list the variables included and their selection basis. In terms of technical details, although the TaqMan typing method was described in detail, the repetition rate or consistency test data in the experiment was not mentioned to support the reliability of the results. This text provides a detailed description of the study design, sample selection, genotyping methods, and statistical analysis in the Materials and Methods section, which reflects scientific rigor. However, the description could be made more comprehensive and more relevant to the research question by adding more information on sample representativeness, the rationale for variable selection, and clarifying the connection between the experimental design and the research objectives.

Validity of the findings

Interpretation of statistical results was insufficient, and although rich statistical data were provided, there was less discussion of the biological significance and possible mechanisms of the results. For example, why is the CYP3A5*3 allele associated with poorer kidney function? What pathways might be involved in its mechanism of action? A brief description should be provided to enhance the scientific interpretability of the results. The logical relationship between significant and non-significant results was not highlighted, and some important indicators (such as eGFR, UACR) became insignificant after adjusting for confounding factors, but this phenomenon was not fully discussed. It is recommended to increase the analysis of the reasons for the changes in results before and after adjustment to highlight the influence of confounding factors. The comparison with the control group is not in-depth enough. The results mainly focus on the genotype distribution and their associations within CKD patients. However, there is insufficient in-depth discussion on the differences in CYP3A5 polymorphisms and renal function between the CKD group and the control group, which may have been overlooked. Advantages of the controlled design of the study. There is a lack of description of the charts. Although some tables (such as Table 1, Table 2, Table 3, etc.) are mentioned, the highlights or core findings of the data in these charts are not explained. This may make it difficult for readers to quickly grasp the key points of the study. The impact of sample size on the results is not discussed, and the sample size may be smaller in some subgroups, possibly leading to increased volatility in the statistical results. It is recommended to add a discussion on the impact of sample size on the stability of the results. This paragraph generally describes the research results comprehensively, and the statistical analysis is detailed and logically clear, which can effectively support the research objectives. However, the interpretation of the results can be further deepened, especially the mechanistic exploration of significant and non-significant results, the analysis of differences with the control group, and the interpretation of key figures. If these contents can be supplemented, it will be more helpful to highlight the scientific significance of the association between CYP3A5 alleles and CKD renal function.

Additional comments

Regarding the discussion section, the mechanism is slightly insufficient. Although it is proposed that the CYP3A5*3 allele may affect renal function by reducing creatinine secretion and lowering eGFR, the specific mechanism of action of CYP3A5 in the kidney is still not in-depth enough. For example, the role of CYP3A5 in other parts of the kidney and its interaction with other renal metabolic pathways have not been fully discussed. The specific effects of this enzyme on renal drug metabolism can be further elaborated to support the research hypothesis. The analysis of the effect on drug metabolism can be more in-depth. The article mentions the relationship between the CYP3A5*3 genotype and drug metabolism, but the metabolic effects of CYP3A5 on other drugs, especially those related to CKD treatment, can be discussed in more detail. For example, in addition to immunosuppressants, are other commonly used CKD treatment drugs (such as antihypertensive drugs, diuretics, etc.) also affected by the CYP3A5*3 allele? This discussion will more comprehensively reflect the impact of genotype on treatment strategies. For further supplementation of cross-study, although the limitations of the study were mentioned, the necessity of interdisciplinary and multicenter studies can be further emphasized, especially when considering the genetic background outside of Thailand. In order to further confirm the impact of CYP3A5 alleles on CKD, it is recommended to conduct multi-ethnic comparative studies to explore how genetic and environmental factors interact to affect renal function. The summary of clinical applications in the conclusion section is relatively concise, and the possibility of individualized treatment and drug dose adjustment is mentioned in the final summary, but more in-depth exploration of how these findings can be actually applied in clinical practice can be carried out. For example, how can genetic screening be integrated into existing CKD treatment guidelines? Are there specific clinical pathways or recommendations? This discussion section is detailed, combining existing research results with the results of this study, and proposes a potential link between the CYP3A5*3 allele and decreased renal function in CKD, especially in terms of drug metabolism, with strong clinical application prospects. However, the mechanism analysis and some detailed discussions can be further supplemented to enhance the understanding of the role of the CYP3A5*3 allele in renal function. In addition, larger-scale and more representative longitudinal studies can be considered in the future to further verify these results.

Reviewer 2 ·

Basic reporting

An interesting study on a biomarker of the progression of chronic kidney disease in patients, which is not easy due to the difficulties that prospective clinical studies always present.
However, it presents a series of problems that I will discuss below.

1- In the Introduction section, the basis for the association between the CYP3A5*3 variant and the progression of kidney disease should be much better justified. For example, in lines 87 and 88 it writes “Previous studies have shown varying results regarding the association between CYP3A5*3/*3 and CKD progression across different ethnicities”, but this is neither described nor referenced.
This section needs to be improved and evidence of the involvement of CYP3A5 nonfunction on renal function, the different hypotheses, needs to be added.
2- In the discussion section, the same thing happens as in the introduction, the results are not adequately discussed. For example, nothing is discussed about the potential involvement of 19-HETE and 20-HETE eicosanoids on the progression of CKD and their metabolism by CYP3A5. For example, the arachidonic acid-derived metabolites 19-HETE and 6β-hydroxycortisol are produced by CYP3A5 enzyme. The CYP3A5*3 polymorphism reduces the formation of 19-HETE and this may lead to an increased availability of 20-HETE which has been shown to increase renal vasoconstriction and peripheral vascular resistance, it has been identified as an independent predictor of CKD progression (Knights et al., 2013; Afshinnia et al., 2018).
This should be included as well as improve the discussion.
3- In Table 1, in the eGFR data of the controls, the range goes from 77.6 to 93.6 ml/min/1.73m2, so according to this data, there would be controls with values within CKD stage 2. How can you explain this? This should be better clarified.

References:
Knights, K.M.; Rowland, A.; Miners, J.O. Renal drug metabolism in humans: The potential for drug-endobiotic interactions involving cytochrome P450 (CYP) and UDP-glucuronosyltransferase (UGT). Br. J. Clin. Pharmacol. 2013, 76, 587–602.
Afshinnia, F.; Zeng, L.; Byun, J.; Wernisch, S.; Deo, R.; Chen, J.; Hamm, L.; Miller, E.R.; Rhee, E.P.; Fischer, M.J.; et al. Elevated lipoxygenase and cytochrome P450 products predict progression of chronic kidney disease. Nephrol. Dial. Transplant. 2018, 35, 303–312

Experimental design

The study design presents no problems.

Validity of the findings

The validity of the results is relative due to the limitations of the study, which the authors themselves include in the text. However, the association shown is interesting for continuing to study about CYP3A5 such as biomarker of progression of chronic kidney disease.

Additional comments

No comments

---

## Round 0.2 · Major Revisions

While I appreciate the authors' thorough and detailed responses to Reviewer 1's comments, which have significantly improved the manuscript, there are still several aspects that require further attention before final acceptance:

1. The relationship between CKD staging and the study objectives (Comment 5) remains insufficiently addressed. Though the authors explained their preference for continuous variables over categorical ones, the manuscript would benefit from a clearer rationale in the introduction or methods section explaining why this approach was chosen to evaluate the relationship between CYP3A5 polymorphism and renal function.

2. The discussion of clinical applications (Comment 15) needs strengthening. The authors mention the potential value of genetic screening for personalized treatment, but provide limited concrete suggestions for how these findings could be integrated into existing CKD treatment guidelines. More specific clinical pathways or recommendations would enhance the translational value of this research.

3. The mechanistic discussion (Comment 12) would benefit from further elaboration. While the authors have added information about CYP3A5's role in the kidney, the molecular mechanisms by which CYP3A5*3 specifically impacts renal function remain somewhat underdeveloped. A more detailed explanation of how this polymorphism influences kidney physiology at the molecular level would strengthen the biological plausibility of the findings.

I believe addressing these remaining issues will further enhance the scientific quality and clinical relevance of this valuable contribution to the literature. I therefore request that the authors submit a revised version addressing these specific points."

Reviewer 2 ·

Basic reporting

The authors adequately answer the questions posed and incorporate the changes and improvements proposed in the first revision.

Experimental design

The authors adequately answer the questions posed and incorporate the changes and improvements proposed in the first revision.

Validity of the findings

The authors adequately answer the questions posed and incorporate the changes and improvements proposed in the first revision.

---

## Round 0.3 · accepted · Accept

Authors have addressed my comments.